# *Notched Belly Grain 4*, a Novel Allele of *Dwarf 11*, Regulates Grain Shape and Seed Germination in Rice (*Oryza sativa* L.)

**DOI:** 10.3390/ijms19124069

**Published:** 2018-12-16

**Authors:** Xiaohong Tong, Yifeng Wang, Aiqun Sun, Babatunde Kazeem Bello, Shen Ni, Jian Zhang

**Affiliations:** State Key Lab of Rice Biology, China National Rice Research Institute, Hangzhou 311400, China; tongxiaohong@caas.cn (X.T.); wangyifeng@caas.cn (Y.W.); 15067150284@163.com (A.S.); tunlapa2k3@yahoo.com (B.K.B.)

**Keywords:** rice, *Oryza sativa* L., notched belly grain, germination, Dwarf 11, brassinosteroids, cytochrome P450

## Abstract

Notched belly grain (NBG) is a type of deformed grain shape that has been associated with inferior appearance and tastes in rice. NBG is coordinated by both environments and genetics. In this study, we report on the first map-based cloning of an NBG gene on chromosome 4, denoted *NBG4*, which is a novel allele of *Dwarf 11* encoding a cytochrome P450 (CYP724B1) involved in brassinosteroid (BR) biosynthesis. A 10-bp deletion in the 7th exon knocked down the level of the *NBG4* transcript and shifted the reading frame of the resulting protein. In addition to the dwarf and clustered panicle as previously reported in the allelic mutants, *nbg4* grains also displayed retarded germination and NBG due to the physical constraint of deformed hulls caused by abnormal hull elongation. *NBG4* is constitutively expressed with the highest level of expression in immature inflorescences. In all, 2294 genes were differentially expressed in *nbg4* and wild-type (WT), and evidence is presented that *NBG4* regulates *OsPPS-2*, *OsPRA2*, *OsYUCCA1*, *sped1-D*, and *Dwarf* that play critical roles in determining plant architecture, panicle development, and seed germination. This study demonstrated that *NBG4* is a key node in the brassinosteroid-mediated regulation of rice grain shape.

## 1. Introduction

Rice (*Oryza sativa* L.) is one of the most important cereals in the world, which feeds over half of the global population. Meanwhile, rice has been used as a model specie in plant molecular biology due to its released genome sequences, ample genetic diversity, and genome co-linearity with other crops [1]. Rice grain shape, including grain length, width, and thickness, is associated with grain size and weight, and has been regarded as an important trait in determining yield and grain appearance quality. Rice grain shape is majorly controlled by the spikelet hull, which sets maximum frame for grains to develop [2]. Up to now, numerous grain shape controlling genes or quantitative trait loci (QTLs), which are involved in G protein signaling, phytohormone signaling, ubiquitin-proteasome pathway, and transcriptional regulation, have been identified and extensively reviewed in several publications [2,3,4].

Notched belly grain (NBG) is a type of defective grain shape, in which the grain belly is notched in a deformed shape [5,6]. Severe extent of notching in the belly may give rise to high chalkiness rate and low grain milling quality, thus making it an unfavorable trait in rice production [7,8]. The formation of NBG is coordinated by genetics as well as environmental cues. On average, approximately 5.63% of rice grains display NBG under natural growth conditions, but the ratio may vary from one variety to another [9]. Nagato et al. (2008) found that NBG was able to be created by blocking the nutrition transportation within the period of 5 to 20 days after pollination, and proposed that the formation of NBG was due to the defects of grain filling [10]. It was also suggested that cold stress and light shading could significantly promote NBG formation [11,12]. In genetics, NBG was thought to be controlled by a single locus or a few loci upon the genetic populations used. For example, Pavithran (1977) mapped an NBG gene *Nk* (*Notched kernel*) in the short arm of chromosome 5, which is tightly linked with *gl1* (*Glabrous hull 1*) [13], whereas *Nk2* (*Notched kernel 2*) was restricted to the chromosome 5 long arm region between marker G1103 and R521 [5]. By scanning an F_2_ population derived from the cross between NJ11 and NBG line L1042, Li et al. (2012) uncovered 11 QTLs, including two major QTLs *qPNG-2* and *qPNG-3*. Despite the progress that has been achieved in mapping the NBG genes, no report is available as yet regarding the cloning of the genes. In this study, we report on the map-based cloning, genetic complementation as well as expression profile of *NBG4*, which is a novel allele of *Dwarf 11* encoding a cytochrome P450 (CYP724B1) involved in brassinosteroid (BR) biosynthesis. *NBG4* controls the formation of NBG by differentially regulating the longitudinal elongation of grain hull and caryopsis.

## 2. Results and Discussion

### 2.1. A31 Is an NBG Mutant

In an effort to clone rice grain shape genes, we identified an NBG mutant A31 by screening a T-DNA insertional mutant population in the background of ZH11 (*Oryza sativa* ssp. Japonica). A31 displayed dwarf, less tillers, compact plant architecture, and fascicled spikelets in “Y” or “W” shape (Figure 1A,B and Appendix A). In A31, there was no change in the grain width, but the grain length was dramatically decreased, thus making the grain more roundish in shape and smaller in size, when compared with ZH11 (Figure 1E,G–I). More interestingly, the majority (92.89 ± 0.24%) of the grains exhibited notched grain shape. The notches uniquely occurred in the grain belly, which is on the far side of the vascular bundle responsible for transporting photosynthesis assimilates into grains (Figure 1E). By dissecting off the hulls of the developing seeds, we observed that the shortened hulls physically blocked the longitudinal elongation of the grains; the relatively more robust growth of the vascular bundle side drove the bent of the grain belly, which finally formed a notched belly in the mature grains (Figure 1C,D). Moreover, when the top of the hulls was removed at three days after pollination, the grain shape of A31 became normal as that of ZH11, although the grain was still shorter than ZH11 (Figure 1F). Notably, although the longitudinal elongations were inhibited in both the hull and grain of A31, the extent apparently varied from one to another. The more severe inhibition on hulls essentially set a physical constraint for the grain elongation, and caused NBG in A31. This phenomenon is consistent with a previous report, in which Takeda and Takahashi found that when the upper part of the hull of NBG genotypes was clipped after anthesis, the caryopsis grew up unrestricted and without notches, and the mature kernel was longer than in the grain with the intact hull. This indicates that, in NBG genotypes, longitudinal growth is unbalanced in the caryopsis with respect to the hull [6]. It also implied that the candidate gene of A31 may mediate the hull and grain development through different regulatory pathways. In addition, we carried out seed germination assays on A31 and ZH11, and found that the germination of A31 was significantly retarded after 24 HAI (hours after imbibitions), when compared with ZH11 (Figure 1J). The application of exogenous BRs obviously rescued the lower germination ability of A31 to a comparable level as ZH11, suggesting that A31 is a BR-deficient mutant (Figure 1J,K). However, the application of Gibberellic acid (GA) did not rescue the germination of A31 to the level as ZH11 (Figure 1L).

### 2.2. Map-Based Cloning of *NBG4*

Because the NBG phenotype was not co-segregated with the T-DNA insertions in the mutants, we adopted a map-based cloning strategy to clone the gene responsible for NBG formation in A31. In an F_2_ population derived from the cross between A31 and TN1 (*Oryza sativa* ssp. Indica), the non-NBG and NBG plants followed a 3:1 ratio, suggesting that NBG in A31 is controlled by a single recessive locus (wild type (WT):NBG = 197:81, χ^2^ = 1.16 < χ^2^_0.05,1_ = 3.84). An F_2_ population containing around 3500 individual plants, including 882 NBG plants, was used for the genetic mapping. First, 136 polymorphous DNA markers which are evenly distributed on the 12 rice chromosomes were used to screen against the two parents and a pooled DNA sample from 24 typical F_2_ NBG plants. This step allowed us to draft map the gene in the long arm of Chr 4, therefore we named the gene *NBG4* (*Notched belly grain 4*) (Figure 2A). Then, 48 individual F_2_ NBG plant DNA samples were screened using the 7 polymorphous DNA markers in this region, and restricted the gene to a region in the genetic distance of 4 cM. Finally, we used 8 polymorphous DNA markers to screen 882 individual F_2_ NBG plant DNA samples, and fine-mapped the gene in a fragment flanked by marker Ind 8 and Ind 14. Sequence analysis showed that this 34-kb region contains 5 open reading frames (Figure 2A).

Sanger sequencing did not detect any mutations, except for a 10-bp deletion in the 7th exon of ORF 5 (*LOC_Os04g39430*), which should have shifted the ORF frame in the downstream of the mutation site, and altered the functions of the resulting protein (Figure 2A). In the mRNA level, *ORF5* transcription in *nbg4* was significantly lower than ZH11 (Figure 2B). Subsequently, we transformed a genomic DNA fragment containing ORF5 and its native promoter into the *nbg4* background, and found that all of the 7 transgenic positive lines were rescued with normal grain shapes, panicle morphology, and plant architecture (Figure 2C,D). Thus, we concluded that *ORF5* is *NBG4*. *NBG4* has been previously reported as *D11* (*Dwarf 11*)/*SG4* (*Small grain 4*)/*CS4* (*Clustered spikelets 4*) [14,15,16]. It encodes a cytochrome P450 (CYP724B1) protein, which has been found as a key enzyme to supply 6-deoxotyphasterol and typhasterol for BR biosynthesis [14]. In accordance with its biochemical functions, the mutants of *D11/SG4/CS4* together with *nbg4* in the current study exhibited typical phenotypes of BR-deficient mutants with abnormal grain size, panicle morphology, and plant architecture. It is also understandable that *nbg4* had a lower seed germination speed, considering that BR generally promotes seed germination. Nevertheless, none of the studies have associated *D11/SG4/CS4* with NBG, suggesting that *NBG4* is a novel allele of *D11/SG4 /CS4*, and different mutation types may give rise to variant phenotypes.

We performed qRT-PCR to examine the mRNA level of *NBG4* in various rice tissues, including root, leaf, inflorescence, and seeds in different growth stages. *NBG4* is constitutively transcribed in all tested tissues (Figure 3A). Its peak level was detected in early inflorescence, while developing seeds had relatively low levels, suggesting that impacts of *NBG4* on the rice hull and seed development are different. During seed germination, the *NBG4* level gradually increased in the germinating embryos until 12 HAI, and then started to go down (Figure 3B).

### 2.3. Downstream Genes Regulated by *NBG4*

We conducted RNA-seq experiments on the 24 HAI germinating embryos of *nbg4* and ZH11 to investigate the downstream genes regulated by *NBG4*. A total of 2294 differentially expressed genes (DEGs) were finally identified, including 1115 up-regulated and 1179 down-regulated in *nbg4* (Appendix A). The DEGs are significantly enriched in the KEGG pathways of “diterpenoid biosynthesis”, “galactose metabolism”, “glutathione metabolism” and “photosynthesis” (Appendix A). To verify the RNA-seq results, we further conducted qRT-PCR analysis on 8 randomly selected DEGs in 24 HAI germinating embryos. It was found that 6 of the genes showed similar inclination of the transcriptional levels as revealed by the RNA-seq, suggesting that our RNA-seq results are highly reliable (Figure 3C). Given the pleiotropical phenotypes in *nbg4*, we examined the transcriptional levels of some reported genes in leaf which have been functionally implicated in seed development, germination, and BR synthesis (Figure 3D). The notched belly of *nbg4* is essentially caused by the severe reduction in hull length. This intriguing phenotype prompted us to test the transcriptional levels of *NSG* (*Non-Stop Glumes*) and *OsMADS1*, the two well-known rice glume development regulatory genes [17,18]. Interestingly, both *NSG* and *OsMADS1* were significantly down-regulated in *nbg4*. In addition, important grain size regulators such as *GW2* (*Grain Width 2*), *GW5* (*Grain Width 5*), *GS6* (*Grain Size 6*), and *qGW8* (*Grain Weight 8*) were investigated [19,20,21,22]. We found that *GS6* was significantly down-regulated in *nbg4*, whereas *GW5* and *qGW8* remained unchanged in the mutant. As a negative regulator of grain width, the transcription level of *GW2* was elevated in *nbg4*, which agrees with the reduced grain width of the mutant with cut hulls (Figure 1H). We also revealed that the mutation of *NBG4* repressed the expression of a couple of plant or panicle architecture genes, such as *OsPPS-2* (*phosphatidyl serine synthase*), *OsPRA2* (*homolog of pea PRA2*), *OsYUCCA1*, *sped1-D* (*shortened pedicels-D*), and *Dwarf*. *sped1-D* has been reported as a pentatricopeptide repeat protein controlling the rice panicle architecture. Dominant mutation of *sped1* resulted in cluster panicles, which is almost identical to the spikelet pattern in *nbg4* [23]. *OsPPS-2* mediates the longitudinal cell elongation in the vegetative tissues. Its mutant showed extremely shortened uppermost internodes and a fully sheathed panicle [24]. Meanwhile, *OsPRA2*, *OsYUCCA1*, and *Dwarf* also regulate rice tissue elongations by participating in GA, auxin, and BR signaling or biosynthesis pathways, implying a complicated interplay of various phytohormones in this BR-deficient mutant *nbg4* [25,26,27].

## 3. Materials and Methods

### 3.1. Plant Materials and Growth Conditions

The recessive mutant *nbg4* was isolated from an M2 population of Zhonghua 11 (ZH11) (*Oryza sativa*, ssp. Japonica) mutagenized by T-DNA insertion. The mutant was crossed with an Indica rice TN1 to generate an F_2_ mapping population. Plants were cultivated in the experimental field of the China National Rice Research Institute (CNRRI) under natural growing conditions for morphological and physiological analysis. Transgenic rice plants were grown in a phytotron under 14 h light (28 ± 2 °C)/10 h dark (25 ± 2 °C) with a relative humidity of 70–85%. The height of mature plants was manually measured. Grain length and width were examined by a seed phenotyping system (Wangsheng, Hangzhou, China).

### 3.2. Map-Based Cloning, Plasmid Construction, and Plant Transformation

Simple sequence repeat and sequence-tagged site markers were derived from (http://www.gramene.org/microsat/ssr.html; Appendix A). The genomic fragments of candidate genes were PCR-amplified, sequenced, and compared with the wild-type sequence for mutation detection. For complementation of the *nbg4* mutant, a 4.46 kb genomic DNA fragment bearing the *NBG4* gene body plus 2 kb promoter and 1 kb 3’ end was PCR-amplified from the ZH11 genomic DNA and cloned into expression vector pCAMBIA2300 (accession number AF234315) to produce the construct p2300-NBG4, which was then introduced into *nbg4* embryonic calli by *Agrobacterium tumefaciens*-mediated transformation according to a previous report [29]. Primer sequences of the 26 polymorphous markers for fine-mapping can be found in Appendix A.

### 3.3. RNA Preparation and RT-PCR Analysis

Total RNA was extracted by using TRIeasyTM Total RNA Extraction Reagent (Yeasen, Shanghai, China) according to the manufacturer’s manual. First strand cDNA was synthesized using M-MLV reverse transcriptase according to the manufacturer’s instructions (Takara, Dalian, China). The expression levels of *NBG4* in different tissues and stages were determined using CFX96 touch real-time PCR detection system (Bio-Rad, Hercules, CA, USA). Expression was assessed by evaluating threshold cycle (CT) values. The relative expression level of tested genes was normalized to ubiquitin gene and calculated by the 2^−ΔΔCT^ method. The experiment was performed in two biological replicates with three technical triplicates of each. Primer sequences are listed in Appendix A.

### 3.4. RNA-seq Analysis

For RNA-seq analysis, germinating embryos at 24 HAI of *nbg4* mutant and ZH11 were manually collected for RNA-seq analysis. RNA samples were extracted using TRIzol according to the manufacturer’s instructions (Yeasen, Shanghai, China). The cDNA library that was constructed by Biomics (Beijing, China) was sequenced using an Illumina HiSeqTM 2500 platform. Gene expression changes between the samples were analyzed by cuffquant and cuffnorm components in cufflinks(2.2.1) softwaree. DEGs were defined as genes with |log_2_Fold change| ≥ 1 and FDR < 0.01 using EBSeq [30]. For GO analysis of RNA-seq data, we used the GO::TermFinder, KOBAS(2.0) to find different expression gene enrichment, and chose *p* < 0.05 as the cutoff for significant GO terms.

## 4. Conclusions

In summary, we characterized a rice NBG mutant, in which the unbalanced growth of floral glumes and grains physically blocked the longitudinal elongation of grains, and finally caused NBG. Seed germination was retarded in the mutant, and could be rescued by exogenous BRs, suggesting that it is BR-deficient. Employing a positional cloning strategy, we identified the causative gene *NBG4*, which is a novel allele of *Dwarf 11* encoding a cytochrome P450 (CYP724B1) involved in brassinosteroid biosynthesis. A 10-bp deletion reduced the transcriptional level of *NBG4* and shifted the reading frame of the resulting protein. *NBG4* is constitutively expressed with the highest level of expression in immature inflorescences. *NBG4* regulates a serial of genes such as *OsPPS-2*, *OsPRA2*, *OsYUCCA1*, *sped1-D*, and *Dwarf* that play critical roles in determining plant architecture, panicle development, and seed germination.

## Figures and Tables

**Figure 1 ijms-19-04069-f001:**
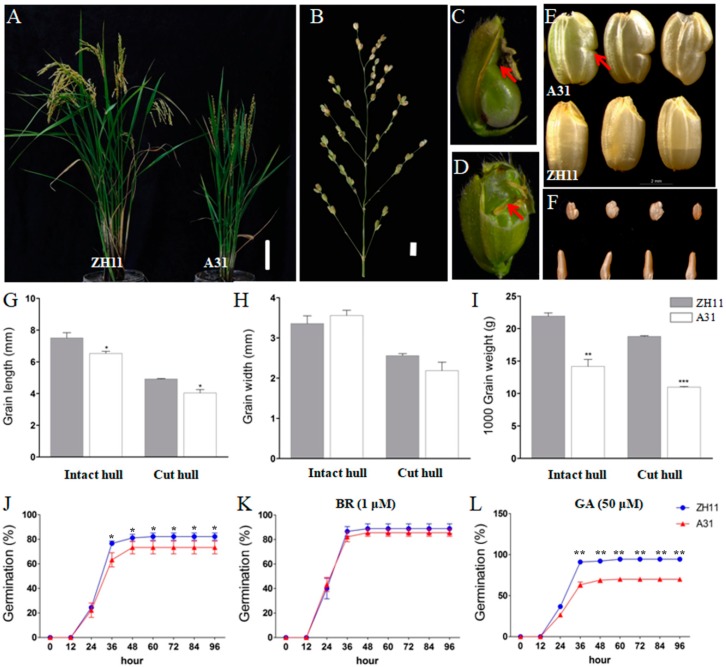
Phenotypes of the rice mutant *nbg4* and wild-type (WT) ZH11. (**A**) Adult ZH11 plant and mutant *nbg4* (A31) plants, bar = 10 cm. (**B**) Clustering panicles of *nbg4* (A31), bar = 1 cm. (**C**,**D**) *nbg4*(A31) developing seeds. Arrows indicate the position of seed notching. (**E**) Grain size of rice plants, bar = 2 mm. Arrows indicate the position of seed notching. (**F**) Grain shapes of *nbg4* with intact hull (**up**) or cut hull (**down**). (**G**–**I**) Average grain length, width, and 1000-grain-weight of the mutant and ZH11, respectively. Numerical values are expressed as the mean; error bars denote one standard deviation of the mean; asterisk denotes significant difference between mutant and WT within each treatment (* *p* < 0.05, ** *p* < 0.01, *** *p* < 0.001; Student’s *t*-test) (hereinafter the same applies) (**J**–**L**) Germination assay of *nbg4* and ZH11 in (**J**) 1/2 Murashige and Skoog (MS) medium, with (**K**) 1 μM brassinosteroid (BR) and (**L**) 50 μM GA (Gibberellic Acid). Asterisk denotes significant difference between A31 and ZH11 at each time point.

**Figure 2 ijms-19-04069-f002:**
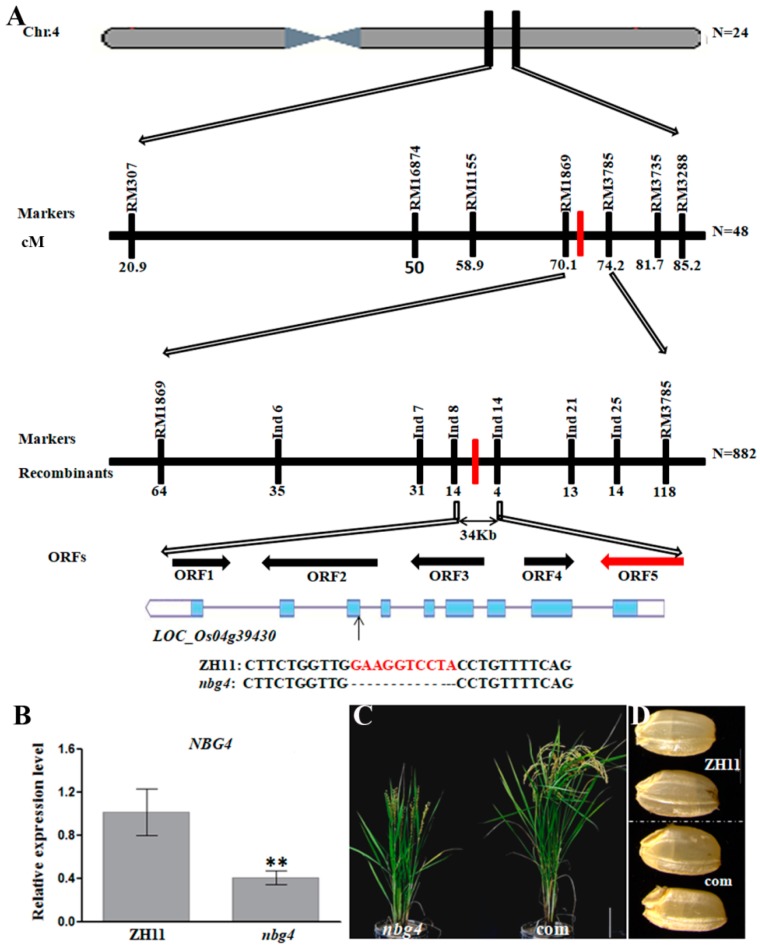
Genetic mapping and candidate gene analysis of *nbg4*. (**A**) Map-based cloning of *NBG4*. The markers for PCR-based mapping are listed in Appendix A. cM: centiMorgans; Recombinants: indicates the recombination between marker and phenotype, the numbers underneath indicate the numbers of recombinant lines. (**B**) Expression of *NBG4* in ZH11 and *nbg4* plants by qRT-PCR analysis. Rice Ubi (OsUBI) was used as an internal control. (** *p* < 0.01; Student’s *t*-test). (**C**) Plant architecture of genetically complemented lines (com), bar = 15 cm. (**D**) Grain shape of complemented plants (com) and ZH11.

**Figure 3 ijms-19-04069-f003:**
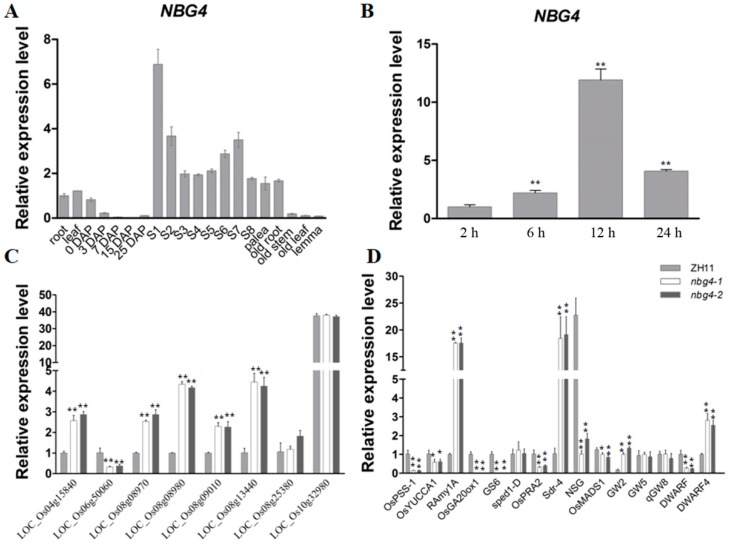
Temporal and spatial profile of *NBG4* and qRT-PCR analysis of *NBG4* downstream genes. (**A**) Transcription levels of *NBG4* in various tissues of ZH11 plants. DAP (Days After Pollination) represents developing seeds in corresponding stages. S1–S8 indicate the spikelets from early differentiation (S1) to fully mature spikelets (S8). Note that *NBG4* has high expression levels in the S1 stage. The level of *NBG4* in root was set at 1.0. (**B**) Expression levels of *NBG4* in germinating embryos, showing that *NBG4* is highly expressed at 12 h after imbibitions. Asterisks indicate significant differences in comparison with 2 h. (**C**) qRT-PCR validation of the transcript levels of DEGs obtained from the RNA-seq results. (**D**) qRT-PCR analysis of transcript levels of well-known genes related to plant and panicle architecture, and grain size in ZH11 and *nbg4*. Rice UBI (Os03g0234200) was used as an internal control [28]. Values are mean ± SE. Asterisks indicate significant differences between ZH11 and *nbg4* (* *p* < 0.05, ** *p* < 0.01; Student’s *t*-test).

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
