# Peer review of "Notched Belly Grain 4*, a Novel Allele of *Dwarf 11*, Regulates Grain Shape and Seed Germination in Rice (*Oryza sativa* L.)"

_ijms, 2018, doi:10.3390/ijms19124069_

Round 1
Reviewer 1 Report
This reviewer found the manuscript to present the results of a study on the structure and function of a rice gene affecting grain shape to have extremely high merit for publication. The premise and experimental plan were well conceived and executed, and the conclusions are solid and important.
The figures developed to present the results and interpretations were very well done and informative. No changes are recommended to improve the graphics.
The reviewer suggests that the title of the paper be changed to:
“Notched belly grain 4, a novel allele of Dwarf 11, regulates grain shape and seed germination in rice (Oryza sativa L.)”
English usage and paragraph and sentence structure, however, needs to be improved significantly before the manuscript is published. Detailed suggestions pertaining to the Abstract are presented below. The reviewer recommends that the authors consult with a colleague with English language skills to bring the paper up to an acceptable standard.
Specific examples of changes needed in the Abstract:
Line 15: NBG expression is modulated by both environmental and genetic factors.; Is there any evidence of genetic x environment interactions?
Lines 16-17: However, no report regarding the cloning of NBG genes is available up to date. “Here, we report the first map-based cloning of the NBG gene NBG4 on chromosome 4, denoted nbg4, …”
Lines 18-19: A 10-bp deletion in the 7th exon knocked down the mRNA level of the NBG4 transcript and shifted the open reading frame of the resulting protein.
Lines 19-22: In addition to the dwarf and clustered panicle as previously reported in allelic mutants in NBG, nbg4 uniquely also displayed retarded germination as well as NBG, majorly due to the physical constraint of deformed hulls caused by the defected abnormal hull longitude elongation.
Lines 22-23: NBG4 nbg4 is constitutively expressed with the highest level of expression predominant transcription in immature early inflorescences.
Lines 23-26: 2294 genes were differentially expressed genes in the NBG mutants and evidence is presented that nbg4 regulates OsPPS-2, OsPRA2, OsYUCCA1, sped1-D 25 and Dwarf that play a role in determining plant architecture, panicle development, and seed germination.
Lines 26-27: This study demonstrated that nbg4 NBG4 is a key node in the BR (define brassinosteroid)-mediated regulation of rice grain shape.
Author Response
Response to reviewer 1:
The reviewer suggests that the title of the paper be changed to: “Notched belly grain 4, a novel allele of Dwarf 11, regulates grain shape and seed germination in rice (Oryza sativa L.)”
Response: we changed the title as you suggested.
English usage and paragraph and sentence structure, however, needs to be improved significantly before the manuscript is published. Detailed suggestions pertaining to the Abstract are presented below. The reviewer recommends that the authors consult with a colleague with English language skills to bring the paper up to an acceptable standard. Specific examples of changes needed in the Abstract:
Response: the language of this MS has been polished by native English speakers. We believe the readability is much better this time.
Line 15: NBG expression is modulated by both environmental and genetic factors.; Is there any evidence of genetic x environment interactions?
Response: we mean here that both environment and genetic factors could modulate NBG. To the best of our knowledge, no report regarding the genetic X environment interactions is available so far.
Lines 16-17: However, no report regarding the cloning of NBG genes is available up to date. “Here, we report the first map-based cloning of the NBG gene NBG4 on chromosome 4, denoted nbg4, …” Lines 18-19: A 10-bp deletion in the 7th exon knocked down the mRNA level of the NBG4 transcript and shifted the open reading frame of the resulting protein. Lines 19-22: In addition to the dwarf and clustered panicle as previously reported in allelic mutants in NBG, nbg4 uniquely also displayed retarded germination as well as NBG, majorly due to the physical constraint of deformed hulls caused by the defected abnormal hull longitude elongation. Lines 22-23: NBG4 nbg4 is constitutively expressed with the highest level of expression predominant transcription in immature early inflorescences. Lines 23-26: 2294 genes were differentially expressed genes in the NBG mutants and evidence is presented that nbg4 regulates OsPPS-2, OsPRA2, OsYUCCA1, sped1-D 25 and Dwarf that play a role in determining plant architecture, panicle development, and seed germination. Lines 26-27: This study demonstrated that nbg4 NBG4 is a key node in the BR (define brassinosteroid)-mediated regulation of rice grain shape.
Response: we corrected the languages according to your suggestive comments. The authors sincerely appreciate your efforts for this paper.
Reviewer 2 Report
The manuscript is interesting and concise. However, it is too much succinct as regards the materials and methods. I therefore request a number of important clarifications in this respect.
Lines 63-64: the authors found “NBG mutant A31 by screening a T-DNA insertional mutant population in the background of ZH11”. Did the authors ascertain if the mutant has a single T-DNA insertion?
Sections 3.1: how many plants the F2 mapping population was constituted of? In Figure 2A, numbers are displayed aside mapping plots of increasing definition; N=882 indicates the number of “recombinant NBG lines” used to map “NBG4 in a 34 kb region containing 5 open reading frames” (lines 107-108). What do, however, N=24 and N=48 indicate? Please, explain.
In Figure 2A, I suppose that ‘CM’ means ‘centiMorgans’, but this should be clarified in the figure caption. Below that, ‘Recombinants’ indicates, I believe, the number of recombinant plants, right? Recombinant with respect to what? I suppose they are recombinants between the marker and the phenotype (and not between nearest markers), but this must be clarified as well.
Section 3.2: there is no real description of the map-based cloning procedure, not even the number of marker used; this is not an acceptable omission. In this respect, on lines 186-187, Table S2 is referred to when mentioning “Simple sequence repeat and sequence-tagged site markers”. Table S2 shows primers for 26 markers used for gene mapping. Really the authors used 26 markers only? Were they enough to identify the region of chr. 4 demarked at the top of Figure 2? I do not think so. Hence, something is missing here.
Section 3.3, does the fact that the qRT-PCR analysis was done in “technical triplicates” mean that there was no biological replication? This is not acceptable, usually. Nevertheless, on line 171, it is said that “similar results were obtained in two independent experiments”. Please, clarify (in the Materials and Methods section).
Section 3.4, how many biological replicates were used for RNA-seq analysis? Which criteria were used for DEG calling? Specifically, which thresholds were adopted for FDR (Benjamini-Hochberg false discovery rate) and fold change?
Line 81, “… germination of A31 wasere significantly retarded, when compared with the ZH11 (Figure 1J)”. Is the difference shown in Figure 1J really significant? What statistical test was done to assess ‘significance’?
Lines 82-83, “Application of exogenous BR, but not by GA, obviously rescued the lower germination ability of A31, suggesting that A31 is a BR-deficient mutant (Figure 1K and L)”. This is not what can be concluded based on Figures 1K and L!! Are, perhaps, these plots erroneously inverted?
Statements based on Figure 1J-L must be supported by a suitable statistical test, because these statements are important, but the observed differences are small.
Some additional minor remarks are reported below.
Line 15: it is mentioned that Notched Belly Grain (NBG) can cause “inferior … tastes in rice”. I am not aware of such an effect, maybe because, apart from mutants, NBG is never extensive enough to affect the taste of the consumed rice. Can a reference be provided for this?
Line 24: “mutants” should be changed to ‘mutant’ or to ‘mutant grains’/ ’mutant spikelets’ or alike.
Line 47: “Nagto” should be corrected to ‘Nagato’.
Line 57, “Here, we report …”: expression analysis and complementation by transformation must be mentioned as well.
Line 61: “discussions” ought to be changed to ‘Discussion’.
Figure 1, “Phenotypes of the of rice mutant nbg4 (A31,right) and wild-type ZH11 (left)”: this should not be the figure title, because it refers to Fig. 1A only.
Figure 1J: there seems to be an incomplete symbols legend in the upper right area of the plot.
Figure 3: of which genotype are the transcription profiles?
Figure 3: the title says this data are from “qRT-PCR analysis”, whereas, on line 168, it is said that these are “(C) Transcript levels of DEGs obtained from the RNA-seq results”. Please clarify.
Lines 133 and 139: I suppose “germinative” could be changed to ‘germinating’.
Line 144: “sever” should be changed to ‘severe’.
Line 157: I surmise “interned” ought to be changed to ‘internodes’.
Line 165: “spekiletes” should be corrected.
Lines 171-172: “Asterisks indicate significant differences between treatments and respective controls at each time point”. I suppose that this is referred to ‘genotypes’, not to “treatments”. What is “each time point” referred to? Please, note that plot B in Figure 3 does not show any asterisk, whereas in plots C and D asterisks are present but there are no time points.
I recommend adding a conclusion statement.
Original studies by Takeda and Takahashi (1970. Genetical studies on rice plant XXXXIL. Varietal differences in the degree of unbalanced growth between caryopsis and floral glumes. Mem. Facul. Agr. Hokkaido Univ. 7:449-453) and/or by Takeda, Ichinohe and Saito (1981. Mechanism of grain notching, and variation for notched grain frequency in rice. Jap. J. Crop Sci. 50:502-508) might be mentioned and discussed.
The manuscript ought to be double-checked because there are many minor editing mistakes.
Author Response
Response to reviewer 2:
Lines 63-64: the authors found “NBG mutant A31 by screening a T-DNA insertional mutant population in the background of ZH11”. Did the authors ascertain if the mutant has a single T-DNA insertion?
Response: we did find T-DNA insertions in the mutant. However, the T-DNA insertions were not co-segregated with the NBG phenotype, thus we conclude that the mutation should be caused by other unknown mutations introduced by the tissue culture process, and decided to clone the candidate gene via a positional cloning strategy.
Sections 3.1: how many plants the F2 mapping population was constituted of? In Figure 2A, numbers are displayed aside mapping plots of increasing definition; N=882 indicates the number of “recombinant NBG lines” used to map “NBG4 in a 34 kb region containing 5 open reading frames” (lines 107-108). What do, however, N=24 and N=48 indicate? Please, explain.
Response: a F2 population containing 3500 individual plants, including 882 NBG plants, was used for the genetic mapping in a process as below: firstly, 136 DNA markers which are evenly distributed on the 12 rice chromosomes were used to screen against two parents and a pooled DNA sample from 24 typical NBG F2 plants, this step allowed us to draft map the gene in the long arm of Chr 4; Secondly, 48 individual F2 NBG plant DNA samples were screened using the 7 DNA markers in this region, and restricted the gene to a more specific region; finally, we used 8 DNA markers to screen the 882 individual F2 NBG plant DNA samples, and fine mapped the gene in a 34-kb fragment. More details are included in the text.
In Figure 2A, I suppose that ‘CM’ means ‘centiMorgans’, but this should be clarified in the figure caption. Below that, ‘Recombinants’ indicates, I believe, the number of recombinant plants, right? Recombinant with respect to what? I suppose they are recombinants between the marker and the phenotype (and not between nearest markers), but this must be clarified as well.
Response: Yes, CM represents centiMorgans, Rcombinants indicates the recombination between the marker and phenotype, the number underneath indicates number of recombinant plants. More detailed descriptions are included in the figure legend. We are sorry for the potential confusions.
Section 3.2: there is no real description of the map-based cloning procedure, not even the number of marker used; this is not an acceptable omission. In this respect, on lines 186-187, Table S2 is referred to when mentioning “Simple sequence repeat and sequence-tagged site markers”. Table S2 shows primers for 26 markers used for gene mapping. Really the authors used 26 markers only? Were they enough to identify the region of chr. 4 demarked at the top of Figure 2? I do not think so. Hence, something is missing here.
Response: as we responded for the question sections 3.1 above, we actually employed 136 polymorphous DNA markers for the draft genetic mapping. We only provided here the sequence of 26 polymorphism markers that were used for fine mapping. More descriptions are added.
Section 3.3, does the fact that the qRT-PCR analysis was done in “technical triplicates” mean that there was no biological replication? This is not acceptable, usually. Nevertheless, on line 171, it is said that “similar results were obtained in two independent experiments”. Please, clarify (in the Materials and Methods section).
Response: we did the qRT-PCR with 2 biological replicates X 3 technical replicates (totally 6 replicates), but we only provided the data of one biological sample with technical triplicates in the first submission. As you suggested, we included the other biological replicates in this version. More details of experiment were also included.
Section 3.4, how many biological replicates were used for RNA-seq analysis? Which criteria were used for DEG calling? Specifically, which thresholds were adopted for FDR (Benjamini-Hochberg false discovery rate) and fold change?
Response: DEGs are defined as genes with |log2Fold change| ≥ 1 and FDR < 0.01 using EBSeq analysis. To reduce the cost, we did not have replicates for RNA-seq in this case. This is because that we only took the RNA-seq results as a clue to find out the potential DEGs and enriched regulatory pathways, and interested DEGs were further validated by more accurate qRT-PCR experiments to assure the reliability of the results.
Line 81, “… germination of A31 wasere significantly retarded, when compared with the ZH11 (Figure 1J)”. Is the difference shown in Figure 1J really significant? What statistical test was done to assess ‘significance’?
Response: students’ t-test analysis confirmed that the differences after 24 hours in figure 1J are significant (P<0.05).< span="">
Lines 82-83, “Application of exogenous BR, but not by GA, obviously rescued the lower germination ability of A31, suggesting that A31 is a BR-deficient mutant (Figure 1K and L)”. This is not what can be concluded based on Figures 1K and L!! Are, perhaps, these plots erroneously inverted?
Response: in figure 1J, the germination of A31 was significantly lower than WT after 24 hours of imbibitions. However, when 1 uM BR was applied, the germination of A31 increased to a comparable level as WT (P>0.05) (Figure 1K), while GA treatment did not alter the germination of A31 to a level similar to WT (Figure 1L). Therefore, we concluded that exogenous BR could rescue the germination of A31, and A31 is possibly a BR-deficient mutant.
Statements based on Figure 1J-L must be supported by a suitable statistical test, because these statements are important, but the observed differences are small.
Response: the statistic differences were confirmed by students’t-test, and also are labeled with asterisks in the resubmitted figure.
Some additional minor remarks are reported below.
Line 15: it is mentioned that Notched Belly Grain (NBG) can cause “inferior … tastes in rice”. I am not aware of such an effect, maybe because, apart from mutants, NBG is never extensive enough to affect the taste of the consumed rice. Can a reference be provided for this?
Response: previous reports (BMC Plant Biology (2017) 17:39; BMC Plant Biology 2014, 14:163) associated the NBG with high rate of chalkiness, which is highly related to the tastes of the grains.
Line 24: “mutants” should be changed to ‘mutant’ or to ‘mutant grains’/ ’mutant spikelets’ or alike. Line 47: “Nagto” should be corrected to ‘Nagato’. Line 57, “Here, we report …”: expression analysis and complementation by transformation must be mentioned as well. Line 61: “discussions” ought to be changed to ‘Discussion’. Figure 1, “Phenotypes of the of rice mutant nbg4 (A31,right) and wild-type ZH11 (left)”: this should not be the figure title, because it refers to Fig. 1A only.
Response: all are corrected!
Figure 1J: there seems to be an incomplete symbols legend in the upper right area of the plot.
Response: this is a typo, we have corrected it.
Figure 3: of which genotype are the transcription profiles?
Response: we used ZH11 for this experiment.
Figure 3: the title says this data are from “qRT-PCR analysis”, whereas, on line 168, it is said that these are “(C) Transcript levels of DEGs obtained from the RNA-seq results”. Please clarify.
Response: we changed the description as “qRT-PCR validation of the transcript levels of DEGs obtained from the RNA-seq results”.
Lines 133 and 139: I suppose “germinative” could be changed to ‘germinating’. Line 144: “sever” should be changed to ‘severe’. Line 157: I surmise “interned” ought to be changed to ‘internodes’.Line 165: “spekiletes” should be corrected.
Response: corrected
Lines 171-172: “Asterisks indicate significant differences between treatments and respective controls at each time point”. I suppose that this is referred to ‘genotypes’, not to “treatments”. What is “each time point” referred to? Please, note that plot B in Figure 3 does not show any asterisk, whereas in plots C and D asterisks are present but there are no time points.
Response: sorry for the typo, we actually mean here the comparison of nbg4 and WT. Statistic results are included in the figure 3B.
I recommend adding a conclusion statement.
Response: conclusion section is added.
Original studies by Takeda and Takahashi (1970. Genetical studies on rice plant XXXXIL. Varietal differences in the degree of unbalanced growth between caryopsis and floral glumes. Mem. Facul. Agr. Hokkaido Univ. 7:449-453) and/or by Takeda, Ichinohe and Saito (1981. Mechanism of grain notching, and variation for notched grain frequency in rice. Jap. J. Crop Sci. 50:502-508) might be mentioned and discussed.
Response: references are cited.
The manuscript ought to be double-checked because there are many minor editing mistakes.
Response: we polished the language and formats by some native English speakers; we believe the readability of this manuscript is much improved. We can not appreciate more for your tremendous efforts and suggestive comments, from which this paper is certainly benefited.
Round 2
Reviewer 1 Report
The manuscript in its current state is acceptable for publication.
Author Response
thank you very much. we appreciate your help for our work.
Reviewer 2 Report
The manuscript is clearer now, but there are still some points to revise.
The fact that the authors found some “T-DNA insertions in the mutant. However, the T-DNA insertions were not co-segregated with the NBG phenotype, thus we conclude that the mutation should be caused by other unknown mutations introduced by the tissue culture process, and decided to clone the candidate gene via a positional cloning strategy” (from the authors’ response to my previous comments) should be briefly mentioned in the manuscript, otherwise it is not clear why a T-DNA insertional mutant population was used.
In my previous comments, I asked “how many biological replicates were used for RNA-seq”? The authors answered that “To reduce the cost, we did not have replicates for RNA-seq in this case. This is because that we only took the RNA-seq results as a clue to find out the potential DEGs and enriched regulatory pathways, and interested DEGs were further validated by more accurate qRT-PCR experiments to assure the reliability of the results”. That’s fine, but it contrasts with the statement that “To verify the RNA-seq results, we further conducted qRT-PCR analysis on 8 randomly selected DEGs” (lines 148-149): it appears that they were not random, but, rather, they were specifically selected as potentially interesting genes. Please, write down plainly what was done in actuality (that is, as described in the response to my previous comments), thus that the readers can fully understand your work. If un-replicated RNA-seq results were used as a clue to find out the potential interesting DEGs, and these were then assayed by accurate qRT-PCR, I believe this can be a reasonable approach, but it must be fully disclosed in the manuscript.
Line 59: “longitude elongation” should be changed to ‘longitudinal elongation’.
Lines 77-79: “This phenomena is consistent with a previous report which suggested that unbalanced growth in floral glumes and caryopsis [6]”. I think this aspect is very important, but I find this sentence not very clear. I’d suggest: ‘This phenomenon was first reported by Takeda and Takahashi [6], who noted that, as the rice caryopsis develops encased in the hull, the size and shape of the former is determined by the latter. Like in the present study, those authors [6] found that when the upper part of the hull of NBG genotypes is clipped after anthesis, the caryopsis grows up unrestricted and without notch, and the mature kernel is longer than in the grain with intact hull. This indicates that, in NBG genotypes, longitudinal growth is unbalanced in the caryopsis with respect to the hull [6]’. In my opinion, a clear description of the studied phenomenon would be greatly appreciated by the readers.
Lines 92-93, “asterisk denotes significant difference between groups”: perhaps it means ‘… significant differences between genotypes within each treatment’?
Line 107, “… restricted the gene to a region in genetic distance of 4 centiMorgans”: I’d say ‘… restricted the gene position into a region of 4 cM’. Note that centiMorgans are normally abbreviated as ‘cM’, not ‘CM’.
Lines 109-110, “… fine mapped the gene in a fragment flanked by marker RM1869 and RM3765”: I think “flanked by marker RM1869 and RM3765” rather refers to “a region in genetic distance of 4 centiMorgans” (lines 107-108). In addition, in Figure 2, the marker on the right is RM3785, not RM3765.
Line 240, “Table S2. Sequences of primers used in this study”: it must be clarified that it is “only provided here the sequence of 26 polymorphism markers that were used for fine mapping” (from the authors’ response to my previous comments).
Lines 212-213: “The relative expression level of tested genes was normalized to ubiquitin gene”. Please, disclose the identity of the reference gene (Os01g0328400? Os02g0161900?). Besides, though “Rice UBI was used as an internal control” (line 182), ubiquitin genes are not stable housekeeping genes in the rice germinating embryo, in my own experience. Hence, expression results shown in Figure 1B could be strongly biased. Please, provide a reference paper for using this reference gene alone.
I still recommend the manuscript is double-checked because there are minor editing mistakes.
Author Response
Response to Reviewer 2:
The fact that the authors found some “T-DNA insertions in the mutant. However, the T-DNA insertions were not co-segregated with the NBG phenotype, thus we conclude that the mutation should be caused by other unknown mutations introduced by the tissue culture process, and decided to clone the candidate gene via a positional cloning strategy” (from the authors’ response to my previous comments) should be briefly mentioned in the manuscript, otherwise it is not clear why a T-DNA insertional mutant population was used.
Response: More descriptions of the mutant are included.
In my previous comments, I asked “how many biological replicates were used for RNA-seq”? The authors answered that “To reduce the cost, we did not have replicates for RNA-seq in this case. This is because that we only took the RNA-seq results as a clue to find out the potential DEGs and enriched regulatory pathways, and interested DEGs were further validated by more accurate qRT-PCR experiments to assure the reliability of the results”. That’s fine, but it contrasts with the statement that “To verify the RNA-seq results, we further conducted qRT-PCR analysis on 8 randomly selected DEGs” (lines 148-149): it appears that they were not random, but, rather, they were specifically selected as potentially interesting genes. Please, write down plainly what was done in actuality (that is, as described in the response to my previous comments), thus that the readers can fully understand your work. If un-replicated RNA-seq results were used as a clue to find out the potential interesting DEGs, and these were then assayed by accurate qRT-PCR, I believe this can be a reasonable approach, but it must be fully disclosed in the manuscript.
Response: This is a misunderstanding. We did picked the 8 DEGs randomly for figure 3C, and the qRT-PCR results were served as a validation of the overall reliability of the RNA-seq. We picked them simply because we previously studied the genes in another project, and had the primers of them in hands. We define it as a random selection here, because most of them are cupin domain containing proteins involving in stress response and disease resistance, which are not related to our research interests in this work.
Line 59: “longitude elongation” should be changed to ‘longitudinal elongation’.
Response: corrected.
Lines 77-79: “This phenomena is consistent with a previous report which suggested that unbalanced growth in floral glumes and caryopsis [6]”. I think this aspect is very important, but I find this sentence not very clear. I’d suggest: ‘This phenomenon was first reported by Takeda and Takahashi [6], who noted that, as the rice caryopsis develops encased in the hull, the size and shape of the former is determined by the latter. Like in the present study, those authors [6] found that when the upper part of the hull of NBG genotypes is clipped after anthesis, the caryopsis grows up unrestricted and without notch, and the mature kernel is longer than in the grain with intact hull. This indicates that, in NBG genotypes, longitudinal growth is unbalanced in the caryopsis with respect to the hull [6]’. In my opinion, a clear description of the studied phenomenon would be greatly appreciated by the readers.
Response: we included this information in.
Lines 92-93, “asterisk denotes significant difference between groups”: perhaps it means ‘… significant differences between genotypes within each treatment’?
Response: corrected.
Line 107, “… restricted the gene to a region in genetic distance of 4 centiMorgans”: I’d say ‘… restricted the gene position into a region of 4 cM’. Note that centiMorgans are normally abbreviated as ‘cM’, not ‘CM’.
Response: corrected.
Lines 109-110, “… fine mapped the gene in a fragment flanked by marker RM1869 and RM3765”: I think “flanked by marker RM1869 and RM3765” rather refers to “a region in genetic distance of 4 centiMorgans” (lines 107-108). In addition, in Figure 2, the marker on the right is RM3785, not RM3765.
Response: it is corrected as marker ind 8 and ind 14.
Line 240, “Table S2. Sequences of primers used in this study”: it must be clarified that it is “only provided here the sequence of 26 polymorphism markers that were used for fine mapping” (from the authors’ response to my previous comments).
Response: we clarified this in the materials and methods section.
Lines 212-213: “The relative expression level of tested genes was normalized to ubiquitin gene”. Please, disclose the identity of the reference gene (Os01g0328400? Os02g0161900?). Besides, though “Rice UBI was used as an internal control” (line 182), ubiquitin genes are not stable housekeeping genes in the rice germinating embryo, in my own experience. Hence, expression results shown in Figure 1B could be strongly biased. Please, provide a reference paper for using this reference gene alone.
Response: the Ubi ref gene ID is Os03g0234200. A reference used it as CK for embryo cDNA was cited. The authors sincerely appreciate your efforts for our paper.